# An In Vitro Anticancer, Antioxidant, and Phytochemical Study on Water Extract of *Kalanchoe daigremontiana* Raym.-Hamet and H. Perrier

**DOI:** 10.3390/molecules27072280

**Published:** 2022-03-31

**Authors:** Justyna Stefanowicz-Hajduk, Anna Hering, Magdalena Gucwa, Katarzyna Sztormowska-Achranowicz, Mariusz Kowalczyk, Agata Soluch, J. Renata Ochocka

**Affiliations:** 1Department of Biology and Pharmaceutical Botany, Medical University of Gdańsk, 80-416 Gdańsk, Poland; anna.hering@gumed.edu.pl (A.H.); magdalena.gucwa@gumed.edu.pl (M.G.); renata@gumed.edu.pl (J.R.O.); 2Department of Pharmacology, Medical University of Gdańsk, 80-204 Gdańsk, Poland; katarzyna.sztormowska-achranowicz@gumed.edu.pl; 3Department of Biochemistry and Crop Quality, Institute of Soil Science and Plant Cultivation, State Research Institute, 24-100 Puławy, Poland; mkowalczyk@iung.pulawy.pl (M.K.); asoluch@iung.pulawy.pl (A.S.)

**Keywords:** cell cycle arrest, HPLC, bufadienolides, human ovarian cancer cells, human keratinocytes, cytostatics, oxidative stress

## Abstract

*Kalanchoe* species are succulents with anti-inflammatory, antioxidant, and analgesic properties, as well as cytotoxic activity. One of the most popular species cultivated in Europe is *Kalanchoe daigremontiana* Raym.-Hamet and H. Perrier. In our study, we analyzed the phytochemical composition of *K. daigremontiana* water extract using UHPLC-QTOF-MS and estimated the cytotoxic activity of the extract on human ovarian cancer SKOV-3 cells by MTT (3-(4,5-dimethylthiazol-2-yl)-2,5-diphenyltetrazolium bromide) assay, flow cytometry, luminometric, and fluorescent microscopy techniques. The expression levels of 92 genes associated with cell death were estimated via real-time PCR. The antioxidant activity was assessed via flow cytometry on human keratinocyte HaCaT cell line. The DPPH (2,2-diphenyl-1-picrylhydrazyl) radical and FRAP (ferric-reducing antioxidant power) assays were also applied. We identified twenty bufadienolide compounds in the water extract and quantified eleven. Bersaldegenin-1,3,5-orthoacetate and bryophyllin A were present in the highest amounts (757.4 ± 18.7 and 573.5 ± 27.2 ng/mg dry weight, respectively). The extract showed significant antiproliferative and cytotoxic activity, induced depolarization of the mitochondrial membrane, and significantly arrested cell cycle in the S and G2/M phases of SKOV-3 cells. Caspases-3, 7, 8, and 9 were not activated during the treatment, which indicated non-apoptotic cell death triggered by the extract. Additionally, the extract increased the level of oxidative stress in the cancer cell line. In keratinocytes treated with menadione, the extract moderately reduced the level of oxidative stress. This antioxidant activity was confirmed by the DPPH and FRAP assays, where the obtained IC_50_ values were 1750 ± 140 and 1271.82 ± 53.25 μg/mL, respectively. The real-time PCR analysis revealed that the extract may induce cell death via TNF receptor (tumor necrosis factor receptor) superfamily members 6 and 10.

## 1. Introduction

The genus *Kalanchoe* (*Bryophyllum*) belongs to the Crassulaceae family and occurs mainly in Asia, Africa, and America. Many species are popular in other regions, including in Europe, due to their pro-health properties. Among the biological activities of *Kalanchoe* plants, the most well-known are their anti-inflammatory, antioxidative, analgesic, antimicrobial, antidiabetic, antimalarial, and antiallergic properties [1,2,3,4,5,6,7,8,9,10]. An anticancer action has also been reported; however, cytotoxic studies on these plants are still being performed with various *Kalanchoe* species and cancer cell lines [4,11,12,13,14,15]. *Kalanchoe daigremontiana* Raym.-Hamet and H. Perrier, also named “mother of thousands”, comes from Madagascar. The cultivation of this species is becoming more widespread due to its medicinal applications and treatments, mainly of leucorrhea, dysmenorrheal, gastric problems, and disorders connected with the central nervous system, such as restlessness, psychic agitation, and anxiety [4,8,16,17,18]. The main secondary metabolites responsible for the biological effects of this plant are bufadienolides (bersaldegenin derivatives, bryophyllin A–C, methyl daigremonate, and daigremontianin) and flavonoids (quercetin, kaempferol, myricetin, and isorhamnetin glycosides) [19,20]. The plant also contains phenolic acids (e.g., gallic, chlorogenic, ferulic, caffeic, *p*-coumaric) and tocopherols [21,22]. Flavonoids are well-known antioxidative plant compounds [23,24]. However, the whole water extract of *K. daigremontiana* that is usually used in medicinal applications has not been evaluated for its antioxidant activity. Until now, a bufadienolide-rich fraction and ethanol extract from this plant have been studied, and the antioxidant efficacy of the fraction was much stronger in comparison with the extract [2,22]. Furthermore, the fraction was tested on blood plasma in vitro, although its antioxidant properties were relatively moderate [2].

Bufadienolides have antiviral, anti-inflammatory, cardiotonic, and toxic activities in in vitro and in vivo models [19,25]. The main mechanism of action in the heart is inhibition of Na^+^/K^+^-ATPase and increases in intracellular Ca^2+^ concentration in cardiac cells [19]. The compounds show a significant effect on heart failure models in rabbits [26]. The cytotoxic activity of bufadienolides in vitro was observed on human leukemia HL-60, lung carcinoma NCI-H460, oral adenosquamous carcinoma Cal-27, lung cancer A549, cervical cancer HeLa, breast cancer MCF-7, ovarian cancer SKOV-3, and melanoma A375 cells [20,27,28,29]. Additionally, the cytotoxic activity of *K. daigremontiana* extracts was evaluated on human lymphoblastic leukemia T [22], human cervical cancer HeLa, ovarian cancer SKOV-3, breast cancer MCF-7, and melanoma A375 cells [20,30]. In our previous study, the water extract of *K. daigremontiana* compared to the ethanol extract showed significant cytotoxic activity, especially on SKOV-3 cells [30]. However, the estimation of cellular factors that play a crucial role in cellular death has not been evaluated. In this study, we conducted a phytochemical analysis of bufadienolides in the water extract of *K. daigremontiana* and estimated the action of this extract on caspase activity, mitochondrial membrane potential, the level of oxidative stress production, and the expression of genes associated with ovarian cancer SKOV-3 cell death. We also evaluated the in vitro antioxidative activity of the extract in DPPH (2,2-diphenyl-1-picrylhydrazyl) and FRAP (ferric-reducing antioxidant power) assays, and next on human keratinocyte HaCaT cells. These new findings presented in this paper allow for future studies on the potential use of *K. daigremontiana* water extract in anticancer therapy.

## 2. Results

### 2.1. Phytochemical Analysis of Bufadienolides in Kalanchoe Daigremontiana Extract

The identification and estimation of the quantities of bufadienolides—the main secondary metabolites in *K. daigremontiana*—were performed in water extracts of the species with HPLC coupled to mass spectrometry. The analysis revealed the presence of twenty steroidal compounds, including bufadienolide, bersaldegenin, daigremontianin, daigredorigenin, and daigremonate derivatives (Table 1).

Furthermore, quantitative analysis showed the contents of eleven compounds, and the amounts of the remaining nine compounds were below the lower limit of detection (Table 1). Bryophyllin A, daigremontianin, and bersaldegenin-1,3,5-orthoacetate were present in the highest amounts in the extract. We also identified bufadienolide compounds 1, 2, 3, 4, and 12, which have not been described in *K. daigremontiana* species.

### 2.2. Kalanchoe Daigremontiana Water Extract Decreased the Viability of SKOV-3 Cells

To estimate the cytotoxic activity of SKOV-3 cells treated with *K. daigremontiana* water extract, we performed an MTT (3-(4,5-dimethylthiazol-2-yl)-2,5-diphenyltetrazolium bromide) assay. The obtained results showed that the percentage of viable cells was 53 ± 3.33% at the extract concentration of 20 μg/mL (Figure 1A), while at the highest used extract concentrations of 100 μg/mL and 200 μg/mL, the results were 41.7 ± 3.34% and 38.3 ± 3.33%, respectively.

We also estimated the effects of *K. daigremontiana* water extract on SKOV-3 cells in the presence of paclitaxel [31,32] and cisplatin [33]. The different ranges of concentrations for both cytostatics in this assay were chosen based on the IC_50_ values of these drugs available in the literature [34]. According to the results of the statistical analysis (ANOVA with post hoc Tukey’s test, *p* < 0.05), we did not observe significant differences between the actions of the extract alone and in the presence of the cytostatic drugs (Figure 1D,E). 

### 2.3. Kalanchoe Daigremontiana Water Extract Induced Cell Death of SKOV-3 Cells

To estimate the viability and induction of cell death, we performed experiments with annexin and 7-aminoactinomycin (7-AAD) [35,36]. The cells were treated with *K. daigremontiana* water extract at concentrations of 40–300 μg/mL for 24 h. The obtained results indicate that the extract induced cell death. The dead cell populations equaled 22.27 ± 0.07%, 33.50 ± 1.16%, 34.14 ± 5.14%, 37.57 ± 0.47%, and 39.85 ± 2.55% for the extract concentrations of 40 μg/mL, 70 μg/mL, 100 μg/mL, 200 μg/mL, and 300 μg/mL, respectively. The early apoptotic populations were not higher than in the control sample. The percentages of late apoptotic cells were 6.53 ± 1.43% and 7.94 ± 0.69% for the higher extract concentrations of 200 μg/mL and 300 μg/mL, respectively (Figure 2).

To confirm the results obtained from flow cytometry, we stained SKOV-3 cells with annexin and propidium iodide (PI) after incubating the cells with *K. daigremontiana* water extract. We mainly observed late apoptotic or necrotic cells with red-stained nuclei and green cellular membranes in samples treated with the extract compared to the untreated cells (Figure 3).

### 2.4. Kalanchoe Daigremontiana Extract Did Not Activate Caspase-3, 7, 8, or 9

The SKOV-3 cells were treated with *K. daigremontiana* water extract at concentrations of 10–300 μg/mL for 2–24 h. The results for the activity levels of caspases-8 and 9 indicate that the extract did not activate the caspases at any tested time point. The relative activity levels of caspases-8 and 9 were lower than 1.0 for the control sample (Figure 4).

The estimations of the activity levels of caspases-3 and 7 were based on a reagent containing a DNA binding dye linked to a DEVD peptide substrate. Active caspases-3 and 7 in the cells cleave this complex and releases the dye, which binds to cellular DNA. As a result increase in fluorescence is observed. Additionally, a dead cell marker (7-AAD) was also included in the assay as an indicator of cell membrane structural integrity and cell death. The living cells did not show increases in fluorescence in the caspase-3 and 7 and 7-AAD parameters (caspases-3 and 7 (-) and 7-AAD (-)). The apoptotic cells exhibited caspase-3 and 7 activity without changes in 7-AAD fluorescence (caspases-3 and 7 (+) and 7-AAD (-)). The late apoptotic and dead cells exhibited increases in fluorescence in the caspase-3 and 7 and 7-AAD parameters (caspases-3 and 7 (+) and 7-AAD (+)), while necrotic cells showed changes only in fluorescence in the 7-AAD parameter (caspases-3 and 7 (-) and 7-AAD (+)). In the case of caspases-3 and 7 in SKOV-3 cells, we did not observe activation after 6 or 24 h of incubation of the cells with the extract. The activity levels of caspases-3 and 7 presented as the population amounts of early apoptotic cells in samples treated with the extract were not higher than in the untreated cells. This indicates that the extract can induce non-apoptotic cell death. We observed significant increases in dead cell populations, the results of which were 13.33 ± 1.21%, 12.04 ± 1.44%, 14.15 ± 2.70%, 30.49 ± 3.29%, and 23.36 ± 4.47% for extract concentrations of 40 μg/mL, 70 μg/mL, 100 μg/mL, 200 μg/mL, and 300 μg/mL, respectively, after 6 h of incubation (Figure 5A). After 24 h of treating the cells with the extract, the percentages of dead cell population were 13.05 ± 1.08%, 23.38 ± 1.02%, 30.67 ± 1.22%, 35.36 ± 1.92%, and 31.38 ± 2.13% for the extract concentrations of 40 μg/mL, 70 μg/mL, 100 μg/mL, 200 μg/mL, and 300 μg/mL, respectively (Figure 5B).

### 2.5. Kalanchoe Daigremontiana Extract Modulated MMP (ΔΨm) in SKOV-3 Cells

*Kalanchoe daigremontiana* water extract was added to SKOV-3 cells, and changes in mitochondrial membrane potential (MMP) were estimated after 6 and 24 h of treating the cells with the extract. The obtained flow cytometry plots showing live, live–depolarized, dead–depolarized, and dead cells were defined according to the cell size index vs. population health profiles of the treated cells in comparison with the untreated control. The results showed that the extract depolarized the membrane of mitochondria and that these changes were significant after both 6 and 24 h. The percentages of depolarized live cells after 6 h were 8.01 ± 0.36%, 11.73 ± 1.74%, 16.39 ± 3.92%, 16.59 ± 1.75%, and 28.98 ± 5.52% for extract concentrations of 40 μg/mL, 70 μg/mL, 100 μg/mL, 200 μg/mL, and 300 μg/mL, respectively (Figure 6A). After 24 h, the percentages of depolarized live cells were 18.46 ± 1.87%, 24.39.± 2.62%, 20.87 ± 1.55%, 35.84 ± 1.25%, and 29.99 ± 1.85% for the extract concentrations of 40 μg/mL, 70 μg/mL, 100 μg/mL, 200 μg/mL, and 300 μg/mL, respectively (Figure 6B).

### 2.6. Kalanchoe Daigremontiana Extract Moderately Increased ROS Production in SKOV-3 Cells

The SKOV-3 cells were incubated with the extract for 24 h. The results showed that the extract induced oxidative stress (ROS) in the cells, and these changes were significant above the extract concentration of 40 μg/mL. The highest percentages of ROS-positive (ROS (+)) cells were for the extract concentrations above 100 μg/mL (Figure 7). The results were 8.43 ± 1.62%, 11.99 ± 2.19%, and 21.16 ± 3.06% for the concentrations of 100 μg/mL, 200 μg/mL, and 300 μg/mL, respectively.

Additionally, we performed experiments with non-cancer HaCaT cells to estimate the antioxidant activity of the extract on this cell line. To induce oxidative stress, the cells were treated with menadione (vitamin K_3_, 2-methyl-1,4-naphthoquinone) at concentrations of 100 μM and 200 μM, and the extract was added to the cells at a concentration range of 40–200 μg/mL. Menadione alone caused cellular oxidative stress at both used concentrations, while the percentages of ROS (+) cells were 25.17 ± 4.33% and 40.15 ± 3.69%, respectively. The obtained results from analysis of the cells incubated both with menadione and the extract indicated that oxidative stress levels decreased compared to those treated with menadione alone. However, the significant effect was only observed in the cells treated with the higher dose of menadione and higher doses of the extract. In this case, the amounts of ROS (+) cells were 40.15 ± 3.69% for menadione alone (200 μM) and 29.41 ± 0.16%, 31.75 ± 0.28%, and 23.75 ± 1.95% for the cells treated with menadione (200 μM) at extract concentrations of 70 μg/mL, 100 μg/mL, and 200 μg/mL, respectively (Figure 8). 

### 2.7. Antioxidant Efficiency of Kalanchoe Daigremontiana Water Extract

To estimate the antiradical efficiency of the water extract of *K. daigremontiana,* we performed the DPPH (2,2-diphenyl-1-picrylhydrazyl) assay and obtained IC_50_ values for this extract and ascorbic acid used as a positive control [37]. The results were 1750 ± 140 μg/mL (Figure 9) and 7.55 ± 0.03 µg/mL, respectively. The IC_50_ values were 1271.82 ± 53.25 μg/mL (Figure 9) and 4.8 ± 0.22 μg/mL in the FRAP assay (ferric-reducing antioxidant power) for the water extract and ascorbic acid [37], respectively. Regarding the FRAP assay results, significance differences among the obtained values were observed only for the extract concentrations of 0.1 mg/mL and 0.2 mg/mL, with ferric-reducing activity levels of 16.5 ± 2.82% and 21.97 ± 3.57%, respectively (ANOVA with post hoc Tukey’s test, *p* < 0.05).

### 2.8. Kalanchoe Daigremontiana Extract Did Not Cause H2A.X Activation in SKOV-3 Cells

To estimate the effects of *K. daigremontiana* water extract on histone H2A.X activation, we performed an experiment with antibody-phospho-specific anti-phospho-histone H2A.X (Ser139)–Alexa Fluor 555. Phosphorylation of histone H2A.X at serine 139 is an important indicator of DNA damage [38]. The level of phospho-histone H2A.X (γH2A.X) increases with the level of double-stranded DNA (dsDNA) damage. Our results indicate that the extract did not cause DNA damage with the activation of H2A.X in contrast to etoposide [39]. The percentage of etoposide treated cells with γH2A.X (named activated) was 85.93 ± 3.26%, whereas for the extract concentrations of 200 μg/mL and 300 µg/mL, we obtained values of 5.9 ± 1% and 8.95 ± 0.93%, respectively (Figure 10). The increasing amount of “non-expressing cells” was due to increase in the amount of cellular debris in the tested cell populations.

### 2.9. Kalanchoe Daigremontiana Extract Induced Cell Cycle Arrest in S and G2/M phases

To estimate the cell cycle arrest of SKOV-3 by *K. daigremontiana* water extract, we performed experiments with flow cytometry. The cells were treated with the extract for 48 h and then stained with reagents from the Muse Cell Cycle Kit. The results indicate that the extract strongly inhibited the cell cycle in S and G2/M phases. The amounts of cells in the S phase were 13.7 ± 0.26%, 15.3 ± 2.12%, 17.37 ± 2.64%, 21.95 ± 0.78%, and 19.5 ± 1.27% for the extract concentrations of 40 μg/mL, 70 μg/mL, 100 μg/mL, 200 μg/mL, and 300 μg/mL, respectively. The amounts of cells in the G2/M phase were 23.05 ± 1.18%, 31.85 ± 0.92%, 39.35 ± 0.49%, 39.97 ± 3.96%, and 36.6 ± 0.99% for the extract concentrations of 40 μg/mL, 70 μg/mL, 100 μg/mL, 200 μg/mL, and 300 μg/mL, respectively (Figure 11).

### 2.10. Kalanchoe Daigremontiana Extract Increased the Expression of Genes Involved in Cell Death

To determine the cell death pathway in SKOV-3 cells treated with *K. daigremontiana* extract, the expression levels of 92 related death genes were tested with a qPCR human apoptosis array [40,41]. The expression levels of seven genes were significantly downregulated (their values were ≤ 0.5-fold lower than the control value), including BAD (Bcl2-associated agonist of cell death), BCL3 (BCL3 transcription coactivator), CASP6 (caspase 6), CASP9 (caspase 9), LRDD (PIDD-p53-induced death domain protein 1), NFKBIZ (NF-kappa-B inhibitor zeta), and TNFSF10 (tumor necrosis factor ligand superfamily member 10). Furthermore, the expression levels of thirteen genes were significantly upregulated and their values were ≥ 2-fold higher than the control value, including BCAP31 (B-cell-receptor-associated protein 31), BCL2A1 (BCL2-related protein A1), BIRC3 (Baculoviral IAP Repeat-Containing Protein 3), BIRC6 (baculoviral IAP repeat-containing protein 6), NOD1 (baculoviral IAP repeat-containing protein 6), CASP1 (Caspase 1), CRADD (CASP2- and RIPK1-domain-containing adaptor with death domain), FAS (tumor necrosis factor receptor superfamily, member 6), LTB (tumor necrosis factor C), NFKB2 (nuclear factor kappa B subunit 2), PEA15 (proliferation and apoptosis adaptor protein 15), RELB (nuclear factor of kappa light polypeptide gene enhancer in B-cells 3), and TNFRSF10 (tumor necrosis factor receptor superfamily, member 10) (Figure 12).

## 3. Discussion

In this study, we estimated the cytotoxic and antioxidative actions of *K. daigremontiana* water extract on human ovarian cancer SKOV-3 cells and keratinocyte HaCaT cells, respectively. Our phytochemical study revealed that this extract has twenty bufadienolide compounds, with the highest contents of daigremontianin, bryophyllin A (bryotoxin C), and bersaldegenin-1,3,5-orthoacetate. In our previous study, we showed that the main bufadienolide, bersaldegenin-1,3,5-orthoacetate, may be responsible for the cytotoxic action of the plant [41]. However, the whole extract, which is the most commonly used option in therapeutic applications, has not been evaluated. The extract showed significant antiproliferative and cytotoxic activities on cancer ovarian cells and strongly inhibited cell cycle in S and G2/M phases. These effects were dose-dependent. Additionally, to estimate the effects of the extract in the presence of cytostatic drugs in SKOV-3 cells, we used two well-known drugs commonly applied in ovarian cancer therapy—paclitaxel and cisplatin. In these experiments, we did not observe an enhanced effect of *K. daigremontiana* water extract in the presence of the cytostatic drugs. The explanation for this result could be due to the different mechanisms of action of the two cytostatics and the analyzed plant extract. The main mechanism of drug-induced cancer cell death is apoptosis, whereby mitochondrial collapse occurs with the release of pro-apoptotic proteins, caspase cascade activation, and DNA damage. Both drugs induce cell cycle arrest in the S phase [31,32,33].

A balance between cycle arrest and proliferation is regulated by various inhibitors and activators of progression in the cell cycle [42,43]. Many plant metabolites can disturb this balance. In our previous study with one of the main bufadienolides of *K. daigremontiana,* bersaldegenin-1,3,5-orthoacetate, we demonstrated that this compound can strongly arrest the cell cycle of HeLa S3 cancer cells in the G2/M phase. The G2/M checkpoint can prevent cells with damaged DNA from undergoing mitosis. In this study, we compared the effects of *K. daigremontiana* water extract with etoposide on the activation of histone H2A.X—a marker of DNA double-strand breaks. The results indicated that the extract did not induce this type of damage as much as etoposide. Only at the highest used concentration of the extract did we observe a slight increase in the number of cells with activated histone H2A.X. It is believed that the cell cycle arrest may be induced even at low levels of DSBs; however, further study is needed to expand on these results, especially that exogenous agents may cause many different types of DNA damage [44].

The water extract of *K. daigremontiana* triggered cell death. In this process, any of the tested caspases were involved in cell death, which indicates that the non-apoptotic process occurs in SKOV-3 cells treated with the water extract. These results are similar to those from our previous study on bersaldegenin-1,3,5-orthoacetate and HeLa cells [41]. This was also confirmed in flow cytometry analysis, whereby mostly dead cell populations significantly increased after incubation with the extract, while the number of early apoptotic cells was not higher than in the control. The similar results obtained by two different assays (with annexin V and caspase-3/7) based on different cell health parameters showed a consistent trend of changes in the cytotoxic effects of *K. daigremontiana*.

Our study also estimated the role of oxidative stress production in SKOV-3 cells treated with the water extract of *K. daigremontiana*. Reactive oxygen species may have an essential role in cell cycle arrest and cell death and can trigger the mitochondria membrane permeability transition (MPT) and loss of membrane potential [45,46,47]. We observed moderate increases in the production of oxidative stress in the ovarian cells, especially at the highest used extract concentrations. The results may suggest that the presence of water extract of *K. daigremontiana* caused the increases in the ROS signaling pathways in SKOV-3 cells under stressful conditions [48,49]. We also demonstrated that depolarization of mitochondrial membrane occurred in SKOV-3 cell death, and this effect was already visible after a few hours of incubation of the cancer cells with the extract. The number of depolarized live cells significantly increased after 24 h of treatment.

Our gene expression analysis revealed that SKOV-3 cell death is induced by death receptor (DR) Fas (CD95) and TNFRSF6/10, which belong to the tumor necrosis factor (TNF) receptor family. This family encompasses CD95, TNFR1, DR3-DR6, nerve growth factor receptor (NGFR), and ectodysplasin receptor (EDAR) [50,51,52,53,54,55]. Recent data indicate that Fas evokes not only apoptosis but also can be involved in non-apoptotic cell death. Death receptors contain the death domain (DD) and transmit the death signal through protein–protein interactions (PPIs). Although the molecular events generating the apoptotic signal induced by DR activation are well described, it is unclear how the receptor activates non-apoptotic signaling pathways [55,56].

Antiradical properties of the water extract of *K. daigremontiana* were also analyzed, and the results showed that this extract lowered the levels of oxidative stress induced by menadione in non-cancer HaCaT cells at higher used concentrations. Kolodziejczyk-Czepas et al., in a study with a bufadienolide-rich fraction of *K. daigremontiana,* indicated that this fraction had moderate antioxidant activity in blood plasma in vitro. Additionally, the DPPH assay showed that the fraction had antiradical activity at a concentration below 100 µg/mL, and the IC_50_ was 21.80 μg/mL [2]. In our study, this effect was observed at water extract concentrations above 1000 µg/mL, and its antioxidant properties (determined in both DPPH and FRAP assays) were much weaker than the bufadienolide-rich fraction. On the other hand, Bogucka-Kocka et al. examined the ethanol extract of *K. daigremontiana* and revealed the antiradical effects on DPPH reduction, with IC_50_ values of the extract ranging between 180 and 1457 µg/mL, depending on the extraction method [22]. This was due to the differences in phytochemical compositions and the concentration of the main metabolites between the fraction and the extracts of *K. daigremontiana*. In addition to cells, the balance between oxidative stress generation and induction of defense mechanisms against ROS in vitro depends on the type of reactive oxygen species, cell line, and laboratory conditions [48,49]. The SKOV-3 cell line proved to be sensitive to the water extract of *K. daigremontiana*, while the same extract indicated only weak antiradical activity on DPPH and iron-reducing properties.

In conclusion, the water extract of *K. daigremontiana* strongly inhibits the cell cycle of human cancer ovarian SKOV-3 cells in the S and G2/M phases and induces cell death connected with activation of transmembrane TNF receptor superfamily members, depolarization of the mitochondrial membrane, and production of oxidative stress. Although the extract has a weak antioxidative effect in vitro, it can, in a moderate way, decrease the levels of menadione-induced oxidative stress in non-cancer cells. According to these results, *K. daigremontiana* water extract has potential in anticancer therapies and should be further studied regarding its mechanisms of action in cancer cells.

## 4. Materials and Methods

### 4.1. Preparation of Kalanchoe Daigremontiana Water Extract

Fresh leaves of *K. daigremontiana* (100.0 g) were obtained from a commercial garden source (Garden Centre Justyna, Gdańsk, Poland). A botanist from the Department of Biology and Pharmaceutical Botany botanically identified the plant. A voucher specimen (No. 21761) was deposited in the Herbarium of the Medical University of Gdańsk (GDMA herbarium). The leaves (100.0 g) were macerated with distilled water (0.5 L) and stirred for 24 h at room temperature. The obtained extract was filtered, concentrated under reduced pressure (40 °C), and lyophilized. The dried extract was dissolved in sterilized distilled water at a concentration of 10.0 mg/mL.

### 4.2. Identification and Quantitative Analyses of Bufadienolides

Dry plant material (10 mg) was suspended in 60% methanol (*v/v*), sonicated for 5 min in an ultrasonic bath, and filtered through a 0.22 μm PVDF membrane in 0.5 mL filtration vials. Analyses were carried out using a UHPLC Thermo Ultimate 3000RS system (Thermo Fisher Scientific, Waltham, MA, USA) equipped with a charged aerosol detector (CAD, Thermo Corona Veo RS, Waltham, MA, USA and coupled to the Q-TOF mass spectrometer (Bruker Impact II HD; Bruker, Billerica, MA, USA). Separations were performed using a Waters HSS column (150 mm × 2.1 mm, 1.8 μm; Waters, Milford, MA, USA). The mobile phases were water (solvent A) and acetonitrile (solvent B), both acidified with 0.1% (*v/v*) formic acid. For qualitative analyses to establish the presence of bufadienolides in the plant material and identify them, gradient elution from 20 to 40% of solvent B was applied with a flow rate of 0.5 mL/min at 50 °C.

Quantitative analyses were performed in isocratic conditions with 23% of solvent B and a 0.4 mL/min flow rate at 40 °C. Following the completion of each analysis, the column was washed with 10 column volumes of 100% solvent B and equilibrated to the starting conditions with 10 column volumes of either 20% or 23% solvent B. The sample injection volume was 7.5 μL.

The eluate from the column was split at a ratio of 3:1 between the charged aerosol detector and the mass spectrometer’s ion source. The following instrumental parameters were used for Q-TOF analysis: capillary voltage, 4.0 kV; nebulizer pressure, 0.7 bar; drying gas flow, 6 l/min; drying gas temperature, 200 °C; ion energy, 4 eV; collision RF, 700.0 Vpp; transfer time, 100.0 μs; pre-pulse storage, 10.0 μs. Positively charged ions were acquired over the *m/z* 100–2000 range with a 5 Hz scanning frequency. MS/MS spectra were obtained using automated data-dependent acquisition, in which two of the most intense precursor ions were fragmented by collision-induced dissociation (CID, Ar collision gas). Collision energies were automatically selected from the pre-defined list based on the *m/z* values of fragmented ions and ramped between 75 and 125% of the selected value. The quadrupole and TOF analyzers’ internal mass calibration was based on the sodium formate clusters injected into 10 mM solution in 50% (*v/v*) 2-propanol to the ion source to 20 µL directly before every analysis.

Tentative identifications of saponins were carried out using high-resolution measurements of patent ions *m/z* ratios with errors not exceeding 5 ppm. Chemical formulas were calculated based on these measurements, and structure–formula relationships were verified using SIRIUS4 software (ver. 4.8.2, Dührkop et al., 2019). Isobaric compounds detected in the analyzed samples required additional identification data from the MS/MS fragmentation spectra. These results were compared with the MS/MS spectra of bufadienolides identified and characterized previously in roots of *K. diagremontiana* [57].

For each tentatively identified saponins, extracted ion chromatograms corresponding to protonated molecules of identified bufadienolides were created with a width of 0.01 Da. Signals from MS and CAD detectors were aligned (−3.6 s delay of CAD vs. MS signal), and for each chromatographic peak detected on the extracted ion chromatograms, a corresponding peak (if present) on the signal from the CAD detector was manually integrated. The CAD response was calibrated within the concentration range of 0.15 to 150.0 μg/mL with a series of dilutions from 1.0 mg/mL solution of kalandaigremoside C or bersaldegenin-1,3,5-orthoactetate. The calibrated ratio between the peak area and internal standard peak area was linear in the utilized range of concentrations. All measurements were performed in triplicate. Bruker Data Analysis version 4.4 SR1 was used in data processing (Bruker Daltonik, Bremen, Germany).

### 4.3. Cell Culture

The human ovarian cancer SKOV-3 cell line and human keratinocyte HaCaT cells were obtained from the American Type Culture Collection (ATCC, Manassas, VA, USA). The SKOV-3 and HaCaT cells were cultured in McCoy’s medium and Dulbecco’s modified Eagle’s medium (DMEM), respectively. Both media were supplemented with 100 units/mL of penicillin, 100 µg/mL of streptomycin, and 10% (*v/v*) fetal bovine serum (FBS) (Merck Millipore, Burlington, MA, USA). The cells were incubated at 37 °C and 5% CO_2_.

### 4.4. MTT Assay

MTT (3-(4,5-dimethylthiazol-2-yl)-2,5-diphenyltetrazolium bromide) assay was used to estimate the antiproliferative activity of *K. daigremontiana* water extract. The SKOV-3 cells were seeded in 96-well plates at a density of 5 × 10^3^ cells/well and treated for 24 h with the *K. daigremontiana* water extract at concentrations of 5.0–200.0 µg/mL. After treatment, the cells were incubated with MTT (0.5 mg/mL; Merck Millipore, Burlington, MA, USA) for 3 h, and then DMSO was used to dissolve formazan crystals. A plate reader (Epoch, BioTek Instruments, Santa Clara, CA, USA) was used to measure the absorbance of the solution. The results (± standard deviation (SD)) were obtained from six repetitions in at least two independent experiments. The data are expressed as percentages of the proliferated cells.

The antiproliferative effect of well-known cytostatics in the presence of the water extract of *K. daigremontiana* was also estimated by MTT assay. Paclitaxel and cisplatin were used in the concentration ranges 0.5–4 nM and 4–48 μM, respectively. The extract was used alone in the concentration range of 5.0–200.0 µg/mL and with cytostatics at concentrations of 2 nM and 25 μM for paclitaxel and cisplatin, respectively. The results (± standard deviation (SD)) were obtained from two independent experiments in four repetitions.

### 4.5. Annexin and 7-AAD Assay

We used Annexin V and a Dead Cell Assay Kit (Merck Millipore, Burlington, MA, USA) to assess the viability of the SKOV-3 cells and induction of apoptosis and necrosis in the population treated with *K. daigremontiana* water extract. The cells were seeded in 12-well plates (1 × 10^5^ cells/well) and incubated with the extract at concentrations of 40–300 µg/mL. After 24 h, the cells were stained with the kit reagents, namely annexin V and 7-AAD (7-aminoactinomycin), and analyzed by flow cytometry (Muse Cell Analyzer, Merck Millipore, Burlington, MA, USA). The experiments were performed in three independent repeats.

### 4.6. Annexin-V-Fluos and Propidium Iodide (PI) Staining

The SKOV-3 cells were seeded in a 12-well plate with coverslips (1 × 10^5^ cells/well) and incubated with *K. daigremontiana* water extract at concentrations of 40.0 and 70.0 µg/mL. After 24 h, the cells on the coverslips were stained with Annexin-V-Fluos Staining Kit (Roche, Bazylea, Switzerland) and observed under the fluorescence microscope.

### 4.7. Caspase-3, 7, 8, and 9 Activity

The cells were seeded in 12-well plates (1 × 10^5^ cells/well) and treated with the extract at concentrations of 40.0–300.0 µg/mL. After 6 and 24 h of treatment, the apoptotic status of the cells (based on caspase-3 and 7 activation), cellular plasma permeabilization, and cell death were estimated. The cells were stained with a fluorescent reagent that contained a DNA binding dye linked to a DEVD peptide substrate. When active caspases-3 and 7 cleave this complex, the dye is released and binds to DNA. A dead cell marker (7-AAD) was also used in the assay. The cells were analyzed with flow cytometry (Muse Cell Analyzer). The experiments were performed in three independent repeats.

The activity levels of caspases-8 and 9 were determined using a Caspase-Glo 8 or 9 Assay Kit (Promega, Madison, WI, USA) and Glomax Multi+ Detection System (Promega, Madison, WI, USA). The cells were seeded in 96-well plates (1 × 10^4^ cells/well), then after 24 h they were incubated with *K. daigremontiana* water extract at concentrations of 10–300 µg/mL for 2–24 h. The experiments were performed in three independent repeats.

### 4.8. Mitochondrial Membrane Potential (MMP)

The SKOV-3 cells were seeded in a 12-well plate (1 × 10^5^ cells/well) and incubated with the extract at concentrations of 40.0–300.0 µg/mL. After 6 and 24 h of exposure, the cells were stained with a Muse MitoPotential Assay Kit (Merck Millipore, Burlington, MA, USA). A Muse Cell Analyzer (Merck Millipore, Burlington, MA, USA) was used to determine the percentages of depolarized live and dead cells. All experiments were independently repeated three times.

### 4.9. Reactive Oxygen Species (ROS) Production

The SKOV-3 cells (1 × 10^5^ cells/well, 12-well plates) were treated with the extract in the concentration range of 40.0–300.0 µg/mL. After 24 h of incubation, the cells were stained with a Muse Oxidative Stress Kit (Merck Millipore, Burlington, MA, USA) and analyzed with a Muse Cell Analyzer (Merck Millipore, Burlington, MA, USA). The experiments were done in three independent repeats.

The HaCaT cells (1 × 10^5^ cells/well, 12-well plates) were treated with menadione (Merck Millipore, Burlington, MA, USA) dissolved in DMSO at concentrations of 100 and 200 μM, and with the extract over a concentration range of 40–200 µg/mL. The concentration of DMSO (control) in wells did not exceed 0.2 % (*v/v*). After 24 h, the cells were harvested and stained with reagents from a Muse Oxidative Stress Kit (Merck Millipore, Burlington, MA, USA) and analyzed with a Muse Cell Analyzer (Merck Millipore, Burlington, MA, USA). The experiments were done in three independent repeats. 

### 4.10. In Vitro DPPH and FRAP Assays

The DPPH radical scavenging assay was performed according to Sakdawattanakul et al. [58]. Briefly, 100 µL volumes of different concentrations of *K. daigremontiana* water extract dissolved in DMSO (0.04–2.5 mg/mL) were mixed with 100 µL of 0.06 mM DPPH methanolic solution and incubated at room temperature in the dark. After 30 min, the absorbance was determined at 510 nm using a 96-well microplate reader (Epoch, BioTek Instruments, Santa Clara, CA, USA). Ascorbic acid was used as the standard. DPPH inhibition was calculated by the following equation:DPPH Inhibition (%) = ((A_control_ − A_extract_)/A_control_) × 100 %(1)

The antioxidant activity levels of the extract and the standard compound were expressed as IC_50_ values (the concentration of the analyzed extract or standard substance that causes a decrease in the non-reduced form of the DPPH radical by 50%) with the program GraFit v.7.0 (Erithacus Software, East Grinstead, UK). The assay was conducted over three independent analyses, with three replicates each.

The reducing ability of *K. daigremontiana* extract was determined with the FRAP test, based on the reduction of Fe^+3^ to Fe^+2^ [46]. The course of the analysis was as follows. Here, 30 μL of serial dilutions of the extract and standard substance (placed in a 96-well plate) were mixed with 170 μL of the freshly prepared reaction mixture (0.3 M acetate buffer: 10 mM TPTZ (2,4,6-tri(2-pyridyl)-s-triazine) in 40 mM HCl: 20 mM FeCl_3_ × 6H_2_O at a ratio of 10:1:1 (*v/v*)). The plate was incubated at room temperature for 20 min, then the absorbance was measured at 593 nm. The percentage of reduced iron ions was read from the calibration curve plotted for ascorbic acid (1–1000 µg/mL). The assay was conducted in three independent analyses, with three replicates each. The IC_50_ value—the concentration of the analyzed extract or standard substance (ascorbic acid) that reduces iron ions by 50%—was calculated using GraFit v.7.0 (Erithacus Software, East Grinstead, UK).

### 4.11. Estimation of H2A.X Activation

The SKOV-3 cells were seeded in a 12-well plate (1 × 10^5^ cells/well) and after 24 h treated with *K. daigremontiana* water extract over a concentration range of 40.0–300.0 µg/mL for 24 h. Etoposide was used as a positive control at a concentration of 10 μM. After 24 h, the cells were stained with an H2A.X Activation Dual Detection Kit (Merck Millipore, Burlington, MA, USA) according to the manufacturer’s instructions and analyzed via flow cytometry (Muse Cell Analyzer, Merck Millipore, Burlington, MA, USA). The experiment was repeated three times.

### 4.12. Cell Cycle Analysis

The SKOV-3 cells were seeded in a 6-well plate (5 × 10^5^ cells/well) and incubated with *K. daigremontiana* water extract over the concentration range of 40.0–300.0 µg/mL for 48 h. After treatment, the cells were prepared using a Muse Cell Cycle Assay Kit (Merck Millipore, Burlington, MA, USA), and the amounts of cells in each phase of the cell cycle were determined using a Muse Cell Analyzer (Merck Millipore, Burlington, MA, USA). The experiment was repeated three times.

### 4.13. Real-Time PCR

The SKOV-3 cells were treated with *K. daigremontiana* extract at a concentration of 70.0 µg/mL for 24 h. Then, the total RNA samples of the cells were isolated using the RNeasy Mini Kit (Qiagen, Venlo, The Netherlands). The concentration of RNA was estimated using Agilent Technologies 4200 TapeStation (Agilent Technologies, Sanata Clara, CA, USA), according to the manufacturer’s protocol, and cDNA synthesis was performed using the Maxima First Strand cDNA Synthesis Kit (Thermo Fisher Scientific, Waltham, MA, USA, USA).

Here, cDNA was applied on TaqMan Array Human Apoptosis Fast 96-well plates (Thermo Fisher Scientific). Each plate contained 92 assays for genes associated with cell death and four assays for control genes. The PCR reactions were performed in a StepOnePlus Real-Time PCR System (Thermo Fisher Scientific, Waltham, Ma, USA). The data were obtained from two independently repeated experiments and analyzed with StepOne software v. 2.3 (Thermo Fisher Scientific, Waltham, MA, USA).

### 4.14. Statistical Analysis

Statistical data were obtained using the STATISTICA 12.0 software package (StatSoft Inc., Tulsa, OK, USA). All data are expressed as mean values ± standard deviation (SD). Student’s *t*-test was used to compare the results with the control sample. The statistical significance was set at *p* < 0.05. In DPPH, FRAP, and MTT assays with cytostatics and the extract, the statistical significance among the results was determined via one-way ANOVA with post hoc Tukey’s test (*p* < 0.05).

## Figures and Tables

**Figure 1 molecules-27-02280-f001:**
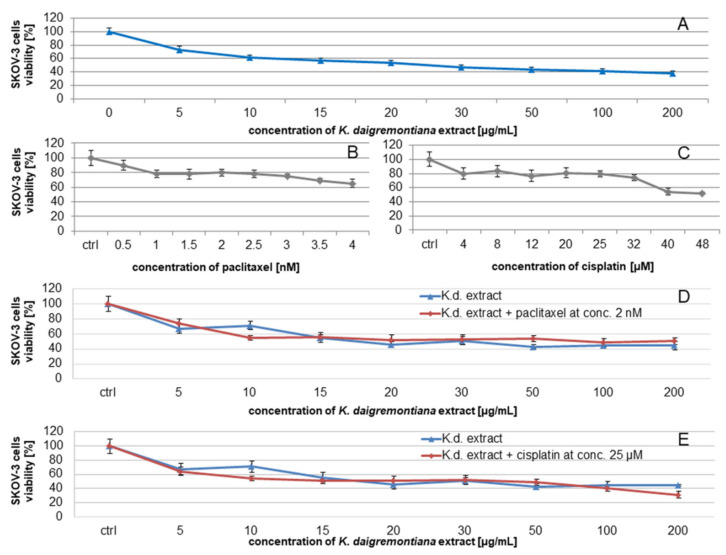
The viability of SKOV-3 cells treated with the water extract of *K. daigremontiana* (**A**) or cytostatics (paclitaxel (**B**) or cisplatin (**C**)) or the extract with the cytostatics (paclitaxel (**D**) or cisplatin (**E**)). The results were obtained via MTT assay (two independent experiments, four repetitions, *n* = 8) after 24 h of incubation of the cells with the extract or cytostatics. Error bars represent standard deviations. ANOVA was performed with post hoc Tukey’s test (*p* < 0.05).

**Figure 2 molecules-27-02280-f002:**
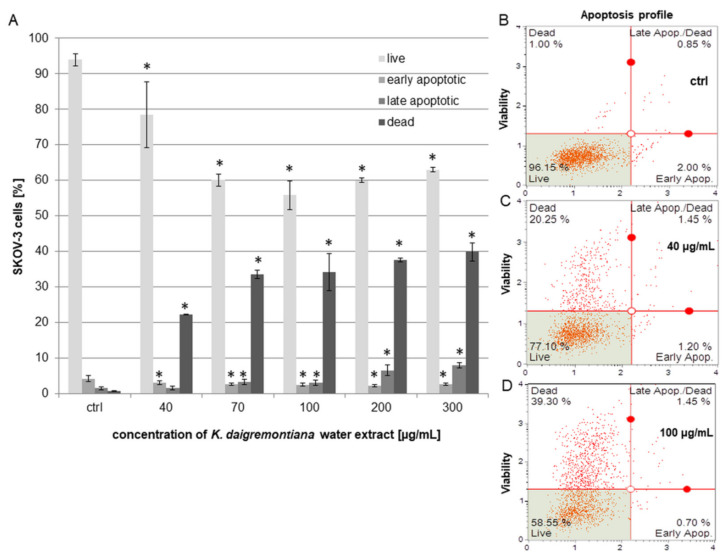
The effects of the water extract of *K. daigremontiana* on the viability of SKOV-3 cells. The estimation of the amount of each population (live, early apoptotic, late apoptotic, and dead cells) was performed with flow cytometry after 24 h of incubation of the cells with the extract (**A**). The values represent the means ± SD of three independent experiments. Error bars indicate standard deviations. Significant differences relative to the control are marked with an asterisk “*” (Student’s *t*-test, *p* < 0.05). The plots show the results after treating the cells with the extract concentrations of 0 μg/mL ((**B**), ctrl), 40 μg/mL (**C**), and 100 μg/mL (**D**).

**Figure 3 molecules-27-02280-f003:**

The SKOV-3 cells processed with annexin-V and propidium iodide after treatment with *K. daigremontiana* water extract. The cells were stained with annexin-V-fluorescein (green) and PI (red) after treating the cells with *K. daigremontiana* extract at concentrations of 0 μg/mL ((**A**)-control), 40 μg/mL (**B**), and 70 μg/mL (**C**). Arrows indicate the late apoptotic or necrotic cells (high-annexin and high-PI staining). The images were captured with a fluorescent microscope at 200× magnification.

**Figure 4 molecules-27-02280-f004:**
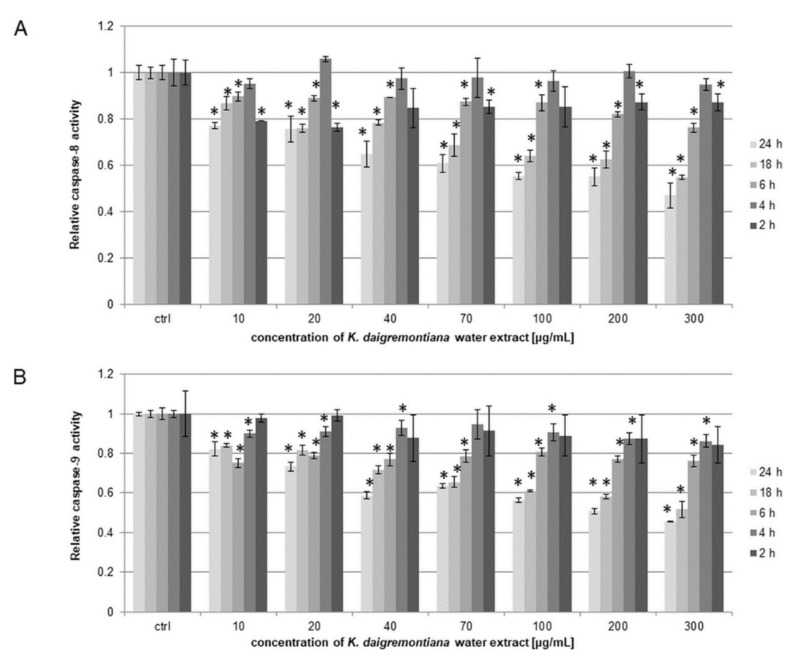
Relative activity levels of caspases-8 (**A**) and 9 (**B**) in SKOV-3 cells treated with *K. daigremontiana* water extract for 2–24 h and estimated with luminometry. The values represent the means ± SD of three independent experiments. Error bars represent standard deviations. Significant differences relative to the control are marked with an asterisk “*” (Student’s *t*-test, *p* < 0.05).

**Figure 5 molecules-27-02280-f005:**
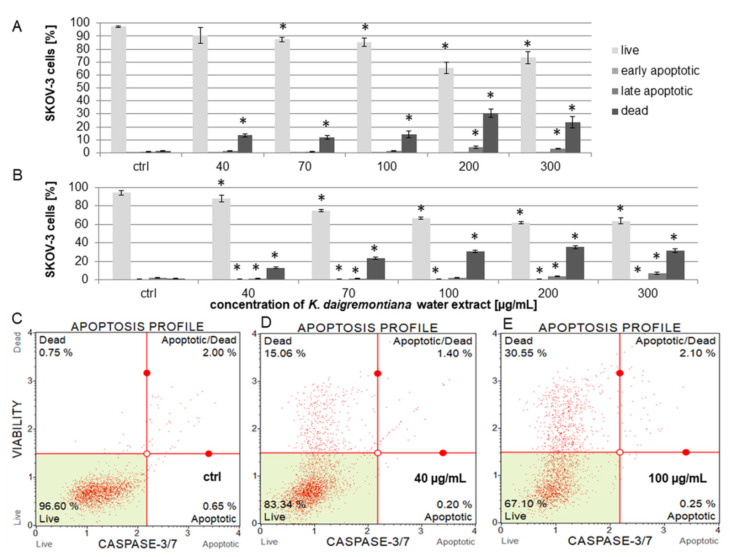
Caspase-3 and 7 activity levels in SKOV-3 cells treated with *K. daigremontiana* water extract for 6 (**A**) and 24 h (**B**), as estimated by flow cytometry and presented as percentages of early, late apoptotic, and dead cells. The values represent the means ± SD of three independent experiments. Error bars represent standard deviations. Significant differences relative to the control are marked with an asterisk “*” (Student’s *t*-test, *p* < 0.05). The plots show the results after 24 h of treating the cells with the extract concentrations of 0 μg/mL ((**C**), ctrl), 40 μg/mL (**D**), and 100 μg/mL (**E**).

**Figure 6 molecules-27-02280-f006:**
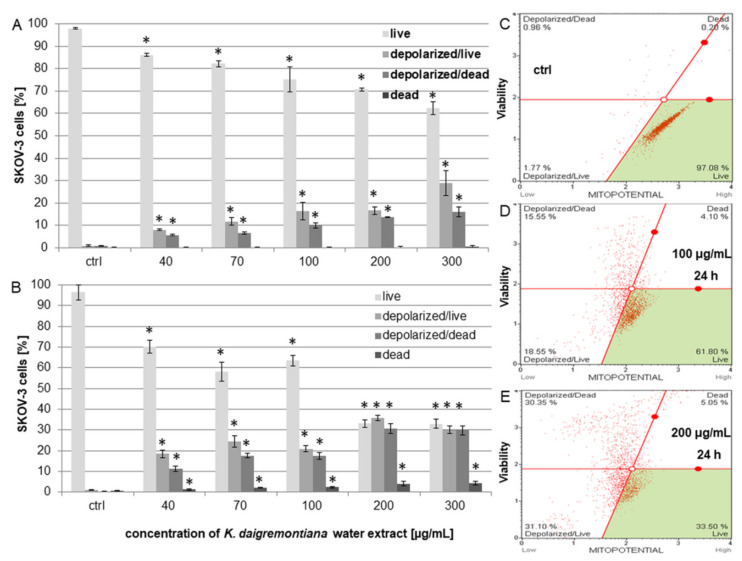
*Kalanchoe daigremontiana* water extract induced decreases in MMP of SKOV-3 cells. The cells were incubated with the extract at concentrations of 40–300 μg/mL for 6 (**A**) and 24 h (**B**) and analyzed via flow cytometry. The values represent the means ± SD of three independent experiments. Error bars represent standard deviations. Significant differences relative to the control are marked with an asterisk “*” (Student’s *t*-test, *p* < 0.05). The plots show the results after 24 h of treating the cells with the extract concentrations of 0 μg/mL ((**C**), ctrl), 100 μg/mL (**D**), and 200 μg/mL (**E**).

**Figure 7 molecules-27-02280-f007:**
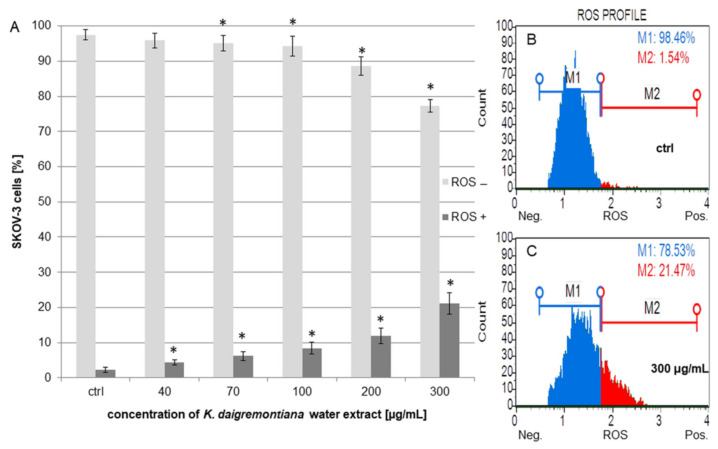
The oxidative stress levels in SKOV-3 cells treated with *K. daigremontiana* water extract for 24 h (**A**). The values represent the means ± SD of three independent experiments. Error bars indicate standard deviations. Significant differences relative to the control are marked with an asterisk “*” (the Student’s *t*-test, *p* < 0.05). The plots show the results after treating the cells with the extract concentrations of 0 μg/mL ((**B**), ctrl) and 300 μg/mL (**C**).

**Figure 8 molecules-27-02280-f008:**
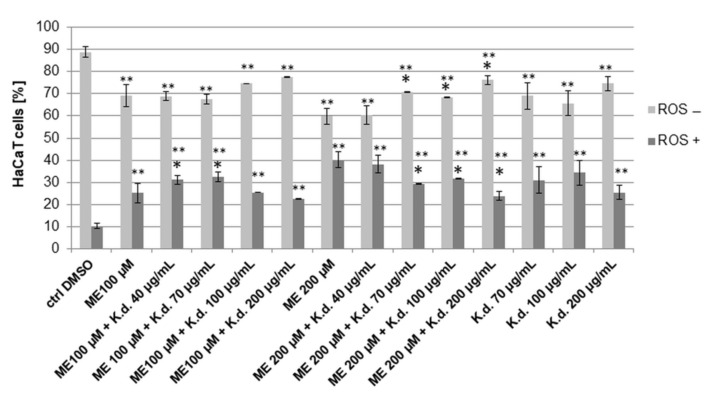
The effects of *K. daigremontiana* water extract on menadione-induced oxidative stress in HaCaT cells. The cells were treated with menadione (ME) and the extract (K.d.) for 24 h and analyzed by flow cytometry. The concentration of DMSO (control sample) was 0.2 % (*v*/*v*). The values represent the means ± SD of three independent experiments. Error bars indicate standard deviations. Significant differences relative to menadione (ME 100 µM or ME 200 µM) or the control are marked with “*” and “**”, respectively (Student’s *t*-test, *p* < 0.05).

**Figure 9 molecules-27-02280-f009:**
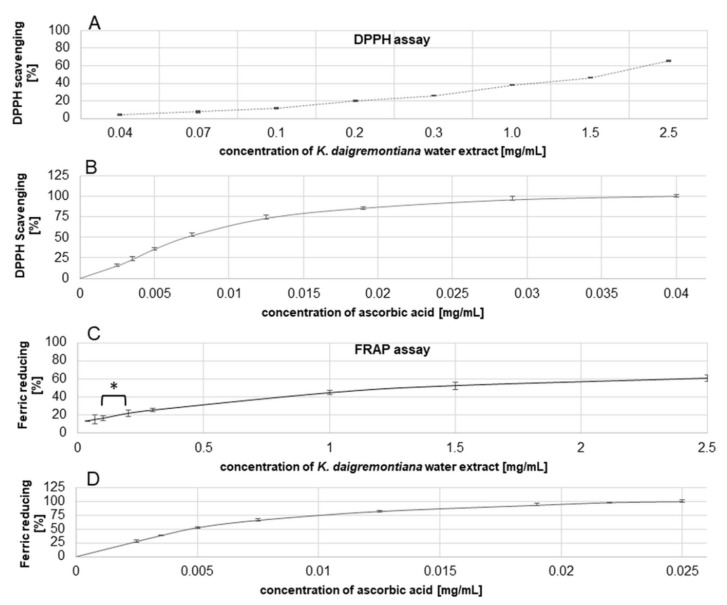
Antioxidant efficiency of *K. daigremontiana* water extract or ascorbic acid estimated in DPPH (**A**,**B**) and FRAP (**C**,**D**) tests. The results were obtained via spectrophotometry from three independent analyses in three replicates (*n* = 9). Error bars indicate standard deviations. The statistical differences among the results were observed only in the FRAP test and are marked with an asterisk “*” (ANOVA with post hoc Tukey’s test, *p* < 0.05).

**Figure 10 molecules-27-02280-f010:**
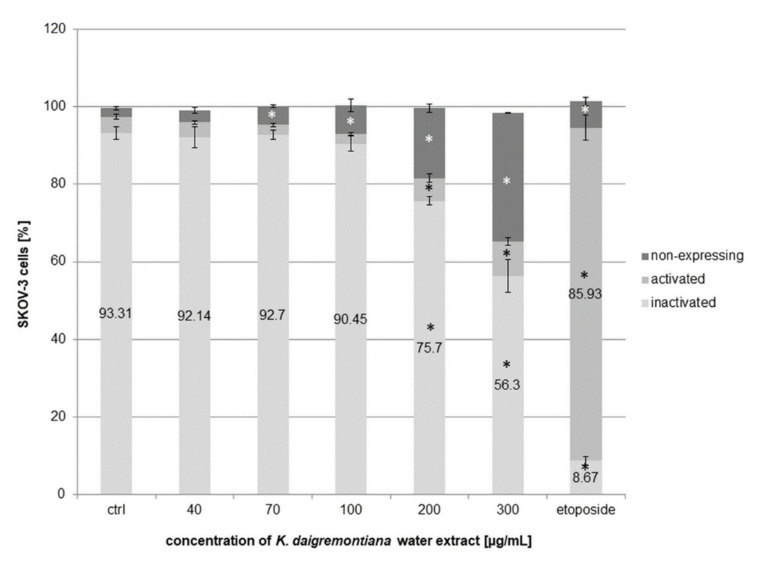
The effects of *K. daigremontiana* water extract on H2A.X activation in SKOV-3 cells. The cells were treated with the extract at concentrations of 40–300 µg/mL and etoposide at a concentration of 10 µM for 24 h. The H2A.X activation was analyzed via flow cytometry. The values represent the means ± SD of three independent experiments. Error bars indicate standard deviations. Significant differences relative to the control are marked with an asterisk “*” (Student’s *t*-test, *p* < 0.05).

**Figure 11 molecules-27-02280-f011:**
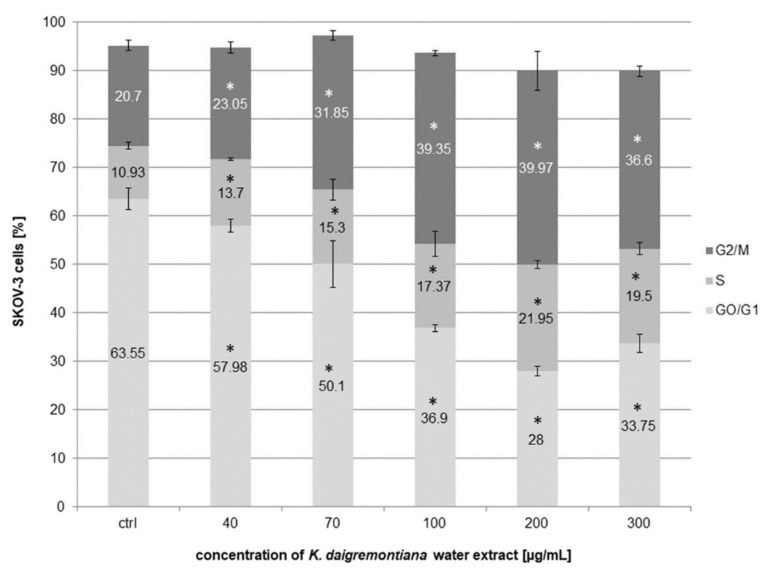
*Kalanchoe daigremontiana* water extract induced cell cycle arrest in S and G2/M phases in SKOV-3 cells. The cells were treated with the extract for 48 h and analyzed by flow cytometry. The values represent the means ± SD of three independent experiments. Error bars indicate standard deviations. Significant differences relative to the control are marked with an asterisk “*” (*p* < 0.05).

**Figure 12 molecules-27-02280-f012:**
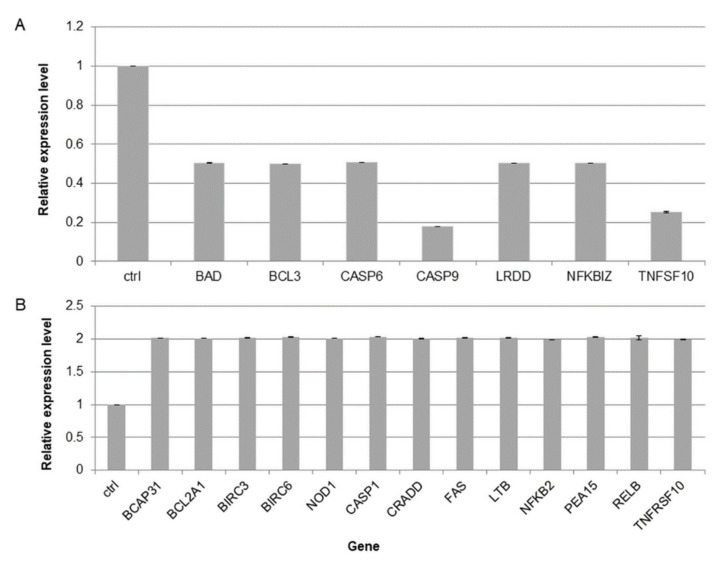
The expression levels of death-associated genes in SKOV-3 cells treated with *K. daigremontiana* water extract at a concentration of 70 μg/mL for 24 h. The expression levels of genes were normalized to the GAPDH endogenous control gene, and their levels are presented as average values ± SD obtained from two independently performed experiments. The results are shown as fold changes under (**A**) or over (**B**) the value of 1.0 (control).

**Table 1 molecules-27-02280-t001:** Identification and content of bufadienolides in *Kalanchoe daigremontiana* water extract using UHPLC-QTOF-MS analysis.

No.	Identification	RT [min]	Meas. *m/z* [M + H]	Ion Formula	mσ	Content [ng/mg d. w.] *
1	tetrahydroxy-bufadienolide-rhamnoside	2.06	581.2954	C_30_H_45_O_11_	17.2	107.2 ± 9.1
2	tetrahydroxy-bufadienolide-dHex isomer 1	2.27	581.2953	C_30_H_45_O_11_	9.3	<LLOD
3	tetrahydroxy-oxo-bufadienolide-acetate isomer 1	2.71	491.2276	C_26_H_35_O_9_	7.1	186.8 ± 16.1
4	tetrahydroxy-bufadienolide-dHex isomer 2	2.94	581.2955	C_30_H_45_O_11_	3.7	<LLOD
5	bersaldegenin-acetate/bryophyllin-C isomer 1	3.68	475.2327	C_26_H_35_O_8_	13.7	35.6 ± 10.5
6	bryophyllin-B/bryotoxin-B isomer 1	4.81	489.2119	C_26_H_33_O_9_	19.2	113.4 ± 12.3
7	daigremontianin isomer 1	4.84	487.1965	C_26_H_31_O_9_	329.8	56.2 ± 8.2
8	bryophyllin-A/bryotoxin-C isomer 1	4.91	473.2174	C_26_H_33_O_8_	5.3	573.5 ± 27.2
9	bryophyllin-B/bryotoxin-B isomer 2	5.05	489.2122	C_26_H_33_O_9_	10.5	<LLOD
10	daigremontianin isomer 2	5.44	487.1967	C_26_H_31_O_9_	14.8	81.4 ± 4.5
11	bryophyllin-A/bryotoxin-C isomer 2	5.73	473.2172	C_26_H_33_O_8_	6.1	<LLOD
12	tetrahydroxy-oxo-bufadienolide-acetate isomer 2	6.06	477.2483	C_26_H_37_O_8_	13.3	<LLOD
13	daigremontianin isomer 3	6.23	487.1967	C_26_H_31_O_9_	8.8	<LLOD
14	bryophyllin-A/bryotoxin-C isomer 2	6.28	473.2175	C_26_H_33_O_8_	31.4	<LLOD
15	bersaldegenin-acetate/bryophyllin-C isomer 2	6.85	475.2332	C_26_H_35_O_8_	3.4	19.2 ± 3.6
16	daigredorigenin-acetate	8.20	461.2537	C_26_H_37_O_7_	7.5	2.0 ± 0.5
17	daigremontianin	8.32	487.1964	C_26_H_31_O_9_	2.6	399.4 ± 19.3
18	methyl-daigremonate isomer 1	10.43	503.2277	C_27_H_35_O_9_	3.0	<LLOD
19	methyl-daigremonate isomer 2	11.94	503.2281	C_27_H_35_O_9_	11.2	<LLOD
20	bersaldegenin-1,3,5-orthoacetate	13.86	457.2224	C_26_H_33_O_7_	6.8	757.4 ± 18.7
**Total content** 2332.1 ± 30						

* Average values of bufadienolide content obtained from three repetitions of the experiment. Note: d.w.—dry weight; dHex—deoxyhexose; < LLOD—below the lower limit of detection.

## Data Availability

The data presented in this study are available in this article.

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
