# Peer review of "An In Vitro Anticancer, Antioxidant, and Phytochemical Study on Water Extract of Kalanchoe daigremontiana Raym.-Hamet and H. Perrier"

_molecules, 2022, doi:10.3390/molecules27072280_

Round 1

Reviewer 1 Report

The research article by Justyna et al., entitled “In vitro anticancer, antioxidant and phytochemical study on water extract of Kalanchoe daigremontiana Raym.-Hamet & H. Perrier” has been reviewed.

The major issue of this manuscript is the novelty and significance parts. The novelty of the manuscript is not very high. The authors did mention and discuss the previous study (Ref. 31 in the manuscript) which may make this manuscript a followed-up study or extended study, but the content of the manuscript showed just simply a small extension of the previous one. In authors’ previous study (Ref. 31), they investigated the effects of bersaldegenin-1,3,5-orthoacetate on DNA damage, cell death and cell cycle arrest in HeLa cells. Though using bersaldegenin-1,3,5-orthoacetate instead of the K. daigremontiana water extract, several parts are similar to the previous one. In current manuscript, the authors investigated the water extract of K. daigremontiana on DNA damage, cell death and cell cycle arrest in SKOV3 cells. However, according to the information of Table 1, bersaldegenin-1,3,5-orthoacetate is one of the dominant compounds in K. daigremontiana water extract, ~32.47% content (757.4 ± 18.7/2332.1 ± 30).

In addition, the structure and coherence of the manuscript are weak. The authors did perform a lot of experiments, however, the organization of the results and the hypothetic model are not properly represented in this manuscript. Also, the abstract lacked a conclusive pat and should be more organized with a focus on the cytotoxic mechanism of K. daigremontiana water extract against SKOV3 cells.

In addition, the following are details that are missing or required revision.

  1. Although the method used (MTT) is ubiquitous to estimate proliferation and viability, it is actually an indirect measure of these parameters. There are many factors that would influence the MTT activity such as metabolic status, mitochondrial activity, and so on. Moreover, MTT couldn’t detect dead cells. Therefore, whether it is appropriate to speak of the results of MTT equal to the antiproliferative activity needs to be considered. (Page 3, line 96.)
  2. Why the authors choose SKOV3 cells, an ovarian cancer cell line, in this manuscript?
  3. Figure 1. The labels and legend of figure 1 are so confusing. I would suggest the authors revise this figure thoroughly. In addition, the statistics is required to see the differences between treatments.
  4. Page 1, line 115. Although the authors said that they did not observe significant changes in the drug's effect in vitro in the presence of the extract, I noticed that there seems difference between kd extract alone and kd+cisplatin in “Figure 1C, group 9”.
  5. Page 4, line 114-117. “We also estimated the effect of daigremontiana water extract on the cytostatic action of paclitaxel and cisplatin in SKOV-3 cells.” please add the references for cytostatic action of paclitaxel and cisplatin. And, the authors should describe the mechanism of these two well-known cytostatic dugs, and compare and discuss the results.
  6. Page 4, line 121. Please add the references for action of annexin and 7-aminoactinomycin (7-AAD).
  7. Page 4, line 121. “40.0-300.0 μg/mL” please correct to “40-300 μg/mL”
  8. Figure 3.: 1.) Please add the information of treatment for 3A, 3B and 3C, respectively.; 2.) Although the authors said that arrows indicate the late apoptotic/necrotic cells (high annexin and high PI staining), I couldn’t see the differences between cell with arrow in 3B or 3C.; 3.) To my knowledge, the methods used annexin V with 7-ADD (Figure 2) or PI (Figure 3) are basically the same, therefore, it’s redundant to show both figure 2 and 3.
  9. Figure 5. 1.) The authors should explain how to translate the caspase-3/7 activity to live/early/late/dead cells in figure 5. ; 2.) 2.) The representative histogram should be shown side by the bar chart. (ex. figure 2).; 3.) The authors should compare and discuss the meaning of results showed in Figures 2 & 5.
  10. Figure 6. 1.) label error: two 100 ug/mL; 2.) the author should describe how they plot the boundary to define the four quadrants.
  11. Figure 8. 1.) Why do the authors choose HaCaT cells to investigate the ROS effect of the extract and the evaluation and correlation with the effects in SKOV3 cells.; 2.) Please add the abbreviations for ME, k.d.; 3.) The total percentages (ROS+ plus ROS-) are not 100% and are even not equal between treatments. The shortened percentage may be due to the dead cells. This would lead to mis-calculate the statistics.
  12. Page 9, line 222. 1.) Please add a reference for ascorbic acid as a positive control. ; 2.) there is no result of ascorbic acid in figure 9.
  13. Figure 9. The authors should explain the statistical significance showed in lower panel.
  14. Page 10, line 234. Please add a reference for “Phosphorylation of histone H2A.X at serine 139 is an important indicator of DNA damage.”
  15. Page 10, line 238. Please add a reference for etoposide (a well-known DNA damage drug)
  16. Figure 10. The authors should discuss why percentage of non-expressing H2A.X is gradually increased upon increasing dose of the water extract.
  17. Figure 11. The overall percentage is not 100%. It may be due to the sub-G1 phase? The authors may show the representative cell cycle profile histograms or add the percentages of sub-G1 in the bar chart.
  18. Page 12, line 263. Please add a reference for qPCR human apoptosis array.
  19. Figure 12. Whether the authors did perform 3 independent experiments for figure 12? Would the authors provide the raw data of Figure 12.
  20. Page 13, line 303. “arrest the cell cycle of cancer cells in the G2/M phase”, suggested to revise to specific HeLa S3 cells.
  21. Page 13, line 328-3. The authors should rephrase the sentence “Our gene expression analysis revealed that SKOV-3 cell death is induced by death receptor (DR) Fas (CD95) and TNFRSF6/10…”.

The authors did observe the upregulation of TNF receptor family (death receptor (DR), Fas (CD95), and TNFRSF6/10) upon treatment with the water extract. However, the authors did not demonstrate the correlation between the water extract-induced death of SKOV3 and the TNF receptor family.

  1. The authors should further investigate whether the daigremontiana water extract-induced cell death of SKOV3 is mediated by TNF receptor family.
  2. In this manuscript, the authors showed that the water extract may induce increases is ROS and also possess antiradical activity. The authors should discuss the possible mechanisms for daigremontiana to exert both ROS-induced and antiradical activities.

Author Response

REVIEWER 1

We would like to thank for critical reading this manuscript and valuable suggestions. We have carefully considered all of the suggestions and made the appropriate corrections.

The research article by Justyna et al., entitled “In vitro anticancer, antioxidant and phytochemical study on water extract of Kalanchoe daigremontiana Raym.-Hamet & H. Perrier” has been reviewed.

The major issue of this manuscript is the novelty and significance parts. The novelty of the manuscript is not very high. The authors did mention and discuss the previous study (Ref. 31 in the manuscript) which may make this manuscript a followed-up study or extended study, but the content of the manuscript showed just simply a small extension of the previous one. In authors’ previous study (Ref. 31), they investigated the effects of bersaldegenin-1,3,5-orthoacetate on DNA damage, cell death and cell cycle arrest in HeLa cells. Though using bersaldegenin-1,3,5-orthoacetate instead of the K. daigremontiana water extract, several parts are similar to the previous one. In current manuscript, the authors investigated the water extract of K. daigremontiana on DNA damage, cell death and cell cycle arrest in SKOV3 cells. However, according to the information of Table 1, bersaldegenin-1,3,5-orthoacetate is one of the dominant compounds in K. daigremontiana water extract, ~32.47% content (757.4 ± 18.7/2332.1 ± 30).

In addition, the structure and coherence of the manuscript are weak. The authors did perform a lot of experiments, however, the organization of the results and the hypothetic model are not properly represented in this manuscript. Also, the abstract lacked a conclusive pat and should be more organized with a focus on the cytotoxic mechanism of K. daigremontiana water extract against SKOV3 cells.

Response: In our manuscript we present the cytotoxic activity of K. daigremontiana water extract which is commonly used in the treatment of many disorders and is more and more popular in many countries. The species has biological activities and among them is cytotoxic action, reported in the literature but not studied in more detail. Thus, we decided to show selected cellular factors that play a role in this cytotoxic activity despite our previous research on bersaldegein-1,3,5-orthoacetate - the main steroidal compound in the complex of the plant bufadienolides. Generally, plant extracts may act in a different way than their main metabolites. Others compounds (in K. daigremontiana there are mainly flavonoids and phenolic acids) may be responsible for the biological effect of the whole extract. To check if this way is different or similar to the main metabolites, we performed research in which we have experiences for many years. Indeed, we agree that the further study should be continued and the role of cellular factors in cell death triggered by the extract should be further developed, however this issue can be the topic of a separate project. Furthermore, we present phytochemical analysis of bufadienolides in the water extract and estimate antioxidant activity – in the case of this plant water extract all these new findings have not been described.

In addition, the following are details that are missing or required revision.

  1. Although the method used (MTT) is ubiquitous to estimate proliferation and viability, it is actually an indirect measure of these parameters. There are many factors that would influence the MTT activity such as metabolic status, mitochondrial activity, and so on. Moreover, MTT couldn’t detect dead cells. Therefore, whether it is appropriate to speak of the results of MTT equal to the antiproliferative activity needs to be considered. (Page 3, line 96.)

Response: The discussion on MTT assay has been around for years. We know that this method in not ideal and has many disadvantages (please see also our work:  Stefanowicz-Hajduk et al. Toxicology Reports 2020, 7:335-344), however it is still used in many studies as the basic method to estimate proliferation and viability of cells treated with compounds/extracts. In Figure 1 we have changed the percentage of cell proliferation to “cell viability” as it is shown in most papers concerning cytotoxic activity of compounds/extracts and also we described the results in the text as percentage of cells viability.

  1. Why the authors choose SKOV3 cells, an ovarian cancer cell line, in this manuscript?

Response: We have chosen SKOV-3 cell line due to the results obtained in our previous study (Justyna Stefanowicz-Hajduk , Anna Hering , Magdalena Gucwa , RafaÅ‚ HaÅ‚asa, Agata Soluch , Mariusz Kowalczyk , Anna Stochmal, Renata Ochocka. Biological activities of leaf extracts from selected Kalanchoe species and their relationship with bufadienolides content, Pharmaceutical Biology, 58:1, 732-740, DOI: 10.1080/13880209.2020.1795208). In that study, we performed screening of cytotoxic activity of different Kalanchoe extract on HeLa, SKOV-3, MCF-7 and A375 cells. In the case of K. daigremontiana extracts, the water extract appeared to be the most active on SKOV-3 cell lines, thus we decided to continue the study with this extract to show its effect on selected cellular factors having an important role in ovarian cell death. This manuscript is the continuation of the topic that we have dealed with for several years.     

  1. Figure 1. The labels and legend of figure 1 are so confusing. I would suggest the authors revise this figure thoroughly. In addition, the statistics is required to see the differences between treatments.

Response: The Figure 1 has been corrected. The statistical analysis (Anova with post hoc Tuckey’s test, p<0.05) did not show differences among the results.

  1. Page 1, line 115. Although the authors said that they did not observe significant changes in the drug's effect in vitro in the presence of the extract, I noticed that there seems difference between kd extract alone and kd+cisplatin in “Figure 1C, group 9”.

Response: We have performed statistical analyses (ANOVA with post hoc Tuckey’s test, p<0.05) and the results did not show differences among the results. This sentence has been added in the text.

  1. Page 4, line 114-117. “We also estimated the effect of daigremontiana water extract on the cytostatic action of paclitaxel and cisplatin in SKOV-3 cells.” please add the references for cytostatic action of paclitaxel and cisplatin. And, the authors should describe the mechanism of these two well-known cytostatic dugs, and compare and discuss the results.

Response: The references have been added in the text. We also shortly discussed the mechanism of action of the cytostatic drugs and the possible explanation of lack of synergistic action of the extract with drugs. However, it was not our goal in this study to compare the mechanism of action of the extract and cytostatics, we show only the effect of this combination on the cells viability.   

  1. Page 4, line 121. Please add the references for action of annexin and 7-aminoactinomycin (7-AAD).

Response: The references have been added in the text.

  1. Page 4, line 121. “40.0-300.0 μg/mL” please correct to “40-300 μg/mL”

Response: We have corrected this.

  1. Figure 3.: 1.) Please add the information of treatment for 3A, 3B and 3C, respectively.; 2.) Although the authors said that arrows indicate the late apoptotic/necrotic cells (high annexin and high PI staining), I couldn’t see the differences between cell with arrow in 3B or 3C.; 3.) To my knowledge, the methods used annexin V with 7-ADD (Figure 2) or PI (Figure 3) are basically the same, therefore, it’s redundant to show both figure 2 and 3.

Response: We have corrected and added necessary information in the figure legend. The Fig. 3  is presented in this manuscript to confirm the effect of the extract on SKOV-3 cells presented in the Fig. 2 and to highlight the tendency in the extract action. We have also improved quality of this picture to show increasing amount of late/necrotic cells in the tested cell populations (treated with the extract at concentrations from 0 to 70 µg/mL).    

  1. Figure 5. 1.) The authors should explain how to translate the caspase-3/7 activity to live/early/late/dead cells in figure 5. ; 2.) 2.) The representative histogram should be shown side by the bar chart. (ex. figure 2).; 3.) The authors should compare and discuss the meaning of results showed in Figures 2 & 5.

Response: We have added this information and placed histograms in the figure. We also compared the results showed in Fig. 2 and 5.

  1. Figure 6. 1.) label error: two 100 ug/mL; 2.) the author should describe how they plot the boundary to define the four quadrants.

Response: The Figure 6 has been corrected. To improve quality we show only the plots from control and samples treated for 24 h (according to the comment of other Reviewer). We also shortly described the rule of boundary settings. 

  1. Figure 8. 1.) Why do the authors choose HaCaT cells to investigate the ROS effect of the extract and the evaluation and correlation with the effects in SKOV3 cells.; 2.) Please add the abbreviations for ME, k.d.; 3.) The total percentages (ROS+ plus ROS-) are not 100% and are even not equal between treatments. The shortened percentage may be due to the dead cells. This would lead to mis-calculate the statistics.

Response: In our study we determined ROS induction in cancer SKOV-3 cell line. Also, we wanted to perform the experiments with non-cancer cells, thus we chose human keratinocytes HaCaT. The abbreviation for ME has been added. The total percentage are not 100%, since in the treatment of the cells with the higher doses of the extract the amount of cellular debrises significantly increases. We performed this flow cytometry analyses according with the manufacturer’s instruction: https://www.luminexcorp.com/muse-oxidative-stress-kit/?wpdmdl=36084).

  1. Page 9, line 222. 1.) Please add a reference for ascorbic acid as a positive control. ; 2.) there is no result of ascorbic acid in figure 9.

Response: We have added the references in the text and results of ascorbic acid in Fig. 9.

  1. Figure 9. The authors should explain the statistical significance showed in lower panel.

Response: We have explained the statistical significance in this Figure.

  1. Page 10, line 234. Please add a reference for “Phosphorylation of histone H2A.X at serine 139 is an important indicator of DNA damage.”

Response: The reference has been added.

  1. Page 10, line 238. Please add a reference for etoposide (a well-known DNA damage drug)

Response: We have added the reference.

  1. Figure 10. The authors should discuss why percentage of non-expressing H2A.X is gradually increased upon increasing dose of the water extract.

Response: We have described this in the Results section.

  1. Figure 11. The overall percentage is not 100%. It may be due to the sub-G1 phase? The authors may show the representative cell cycle profile histograms or add the percentages of sub-G1 in the bar chart.

Response: We have added histograms in the Fig. 11.

  1. Page 12, line 263. Please add a reference for qPCR human apoptosis array.

Response: The references have been added in the text.

  1. Figure 12. Whether the authors did perform 3 independent experiments for figure 12? Would the authors provide the raw data of Figure 12.

Response: We performed two independently experiments. Below we provide the raw data from RT-PCR analysis.

Control

Sample K. daigremontiana extract

the results of CT (average values from two experiments)

the results of CT (average values from two experiments)

Target Name

CÑ‚

Target Name

CT

18S

15,47148991

18S

14,96766

GAPDH

23,97251987

GAPDH

23,98244

HPRT1

27,96638012

HPRT1

26,95391

GUSB

28,93729496

GUSB

28,95425

BIRC2

28,48172092

BIRC2

27,96991

APAF1

31,48576641

APAF1

31,98786

BAD

33,9848938

BAD

34,98297

BAK1

30,46984291

BAK1

29,98886

BAX

29,9771595

BAX

30,47644

BBC3

34,47091103

BBC3

33,98269

BCAP31

27,97041225

BCAP31

26,96907

BCL10

31,44435787

BCL10

31,45245

BCL2

33,96666336

BCL2

33,96705

BCL2A1

36,93249321

BCL2A1

35,93728

BCL2L1

30,49832058

BCL2L1

30,49552

BCL2L11

33,94406319

BCL2L11

33,44438

BCL2L13

30,45150948

BCL2L13

30,94988

BCL2L2

32,47553539

BCL2L2

31,97751

BCL3

29,96706486

BCL3

30,98421

BID

33,48080635

BID

32,98712

BIK

35,46869278

BIK

35,95736

NAIP

36,46675682

NAIP

36,9812

BIRC3

31,95112514

BIRC3

30,94749

XIAP

31,4533577

XIAP

30,95033

BIRC5

28,48004055

BIRC5

28,97497

BIRC6

29,95993233

BIRC6

28,94761

BNIP3

27,94617748

BNIP3

27,9297

BNIP3L

26,95663071

BNIP3L

26,96047

BOK

33,48851395

BOK

33,96825

NOD1

35,94499779

NOD1

34,94915

CARD6

33,97440529

CARD6

33,972

CASP1

32,94318962

CASP1

31,93044

CASP10

32,94798088

CASP10

33,44522

CASP2

31,46324062

CASP2

31,96295

CASP3

28,94543266

CASP3

28,94352

CASP4

29,45704937

CASP4

28,46916

CASP6

29,94808102

CASP6

30,94056

CASP7

30,96648788

CASP7

30,96974

CASP8

30,45553303

CASP8

29,94928

CASP8AP2

31,93758011

CASP8AP2

31,94632

CASP9

33,51523399

CASP9

36,00907

CFLAR

29,958745

CFLAR

29,95956

CHUK

35,95380974

CHUK

35,44851

CRADD

32,98007393

CRADD

31,98581

DAPK1

32,96313667

DAPK1

32,96502

DEDD

29,96847248

DEDD

29,96998

DEDD2

33,4739151

DEDD2

33,97695

DIABLO

29,95250988

DIABLO

29,96319

IFT57

29,4632082

IFT57

28,95826

FADD

32,48779869

FADD

33,47309

FAS

32,9632988

FAS

31,96275

HIP1

34,4681015

HIP1

34,97891

HTRA2

31,96972752

HTRA2

31,97581

IKBKB

31,4405098

IKBKB

31,97578

IKBKE

33,4475174

IKBKE

33,93613

IKBKG

30,96337509

IKBKG

30,96002

LRDD

34,97124481

LRDD

35,97417

LTA

36,47193718

LTA

36,43986

LTB

36,97594833

LTB

35,97554

MCL1

29,47623634

MCL1

28,48695

NLRT1

34,48204041

NLRT1

33,98666

NFKB1

29,96063805

NFKB1

29,47987

NFKB2

30,9777422

NFKB2

29,9961

NFKBIA

29,47056103

NFKBIA

28,96069

NFKBIB

33,46644402

NFKBIB

32,98025

NFBIE

34,96891403

NFBIE

34,46626

NFKBIZ

28,9449234

NFKBIZ

29,94668

PA15

27,97576618

PA15

26,96478

PMAIP1

28,98020077

PMAIP1

27,47952

REL

30,91921806

REL

30,94453

RELA

29,45736122

RELA

28,96925

RELB

33,98875999

RELB

32,98657

RIPK1

30,97151279

RIPK1

31,47114

RIPK2

29,44279003

RIPK2

28,93424

TBK1

29,94403172

TBK1

29,96714

TNF

32,98012924

TNF

32,48788

TNFRSF10

32,96786308

TNFRSF10

31,98492

TNFRSF10B

30,46632099

TNFRSF10B

29,47158

TNFRSF1A

28,47664165

TNFRSF1A

28,47233

TNFRSF1B

36,9646492

TNFRSF1B

37,00249

TNFRSF21

29,9684391

TNFRSF21

29,96752

TNFRSF25

32,98087502

TNFRSF25

33,48155

TNFSF10

30,93126965

TNFSF10

32,93324

TRADD

33,47594643

TRADD

33,95331

  1. Page 13, line 303. “arrest the cell cycle of cancer cells in the G2/M phase”, suggested to revise to specific HeLa S3 cells.

Response: This information has been added in the text.

  1. Page 13, line 328-3. The authors should rephrase the sentence “Our gene expression analysis revealed that SKOV-3 cell death is induced by death receptor (DR) Fas (CD95) and TNFRSF6/10…”.

The authors did observe the upregulation of TNF receptor family (death receptor (DR), Fas (CD95), and TNFRSF6/10) upon treatment with the water extract. However, the authors did not demonstrate the correlation between the water extract-induced death of SKOV3 and the TNF receptor family.

  1. The authors should further investigate whether the daigremontianawater extract-induced cell death of SKOV3 is mediated by TNF receptor family.

Response to 21 and 22: In our study we tested a panel of genes and observed that the extract increased the expression level of the genes that belong to the TNF receptor family. Based on these RT-PCR results, we show in this manuscript only the possible beginning of induction of cells death after the extract treatment. In Discussion section we present that molecular mechanism connected with TNF receptor and non-apoptotic death is unclear. We agree that this result should be evaluated, however this important and extensive topic can be estimated in the next research project.

  1. In this manuscript, the authors showed that the water extract may induce increases is ROS and also possess antiradical activity. The authors should discuss the possible mechanisms for daigremontianato exert both ROS-induced and antiradical activities.

Response: We have discussed this in the Discussion section.

We thank the Reviewer for all suggestions and hope that the revised manuscript is now appropriate for publication.

Sincerely,

Justyna Stefanowicz-Hajduk, Ph.D.

Assistant Professor,

Department of Biology and Pharmaceutical Botany,

Medical University of Gdansk, Gdansk, Poland

Reviewer 2 Report

In this manuscript titled “In vitro anticancer, antioxidant and phytochemical study on 2 water extract of Kalanchoe daigremontiana Raym.-Hamet & H. 3 Perrier” is reported the phytochemical composition of K. daigremontiana water extract by the UHPLC-QTOF-MS and estimated cytotoxic activity of the extract  on human ovarian cancer SKOV-3 cells by MTT (3-(4,5-dimethylthiazol-2-yl)-2,5-diphenyltetrazo lium bromide) assay, flow cytometry, luminometric, and fluorescent microscopy techniques. The antioxidant activity was assessed with flow cytometry on human keratinocytes HaCaT cell line. The DPPH  (2,2-diphenyl-1-picrylhydrazyl) radical and FRAP (Ferric Reducing Antioxidant Power) assays  were also applied as conferm of antioxidant activity.

The subject is suitable for publication in the journal. However, some changes should be introduced.

-Lines 81: rewrite the sentence according to the right order of instrument. The analysis with HPLC coupled to mass spectrometry.

-In Table 1, some compounds are reported as <LOQ. In the manuscript is not reported the LOQ an LOD of the method of UHPLC-QTOF-MS analysis.  Where is reported the validation of the methods?

-Figures  2, 6, and 7 need to be redone. The quality of the images even by computer is not

optimal.

Author Response

REVIEWER 2

We would like to thank for critical reading this manuscript and valuable suggestions. We have carefully considered all of the suggestions and made the appropriate corrections.

In this manuscript titled “In vitro anticancer, antioxidant and phytochemical study on 2 water extract of Kalanchoe daigremontiana Raym.-Hamet & H. 3 Perrier” is reported the phytochemical composition of K. daigremontiana water extract by the UHPLC-QTOF-MS and estimated cytotoxic activity of the extract  on human ovarian cancer SKOV-3 cells by MTT (3-(4,5-dimethylthiazol-2-yl)-2,5-diphenyltetrazo lium bromide) assay, flow cytometry, luminometric, and fluorescent microscopy techniques. The antioxidant activity was assessed with flow cytometry on human keratinocytes HaCaT cell line. The DPPH  (2,2-diphenyl-1-picrylhydrazyl) radical and FRAP (Ferric Reducing Antioxidant Power) assays  were also applied as conferm of antioxidant activity.

The subject is suitable for publication in the journal. However, some changes should be introduced.

-Lines 81: rewrite the sentence according to the right order of instrument. The analysis with HPLC coupled to mass spectrometry.

Response: We have corrected the sentence.

-In Table 1, some compounds are reported as <LOQ. In the manuscript is not reported the LOQ an LOD of the method of UHPLC-QTOF-MS analysis.  Where is reported the validation of the methods?

Response: The method was not yet fully validated. Thus, there is no report on validation results included with the manuscript. We calculated the limits of quantitation according to the ICH guidelines based on the standard deviation of the response (LOQ=3.3*a/s, where a is the slope of calibration curve and s is the response SD), obtained from the residual standard deviation of the regression line.

-Figures  2, 6, and 7 need to be redone. The quality of the images even by computer is not

optimal.

Response: The figures have been redone. In Fig. 2 we have enlarged the plots, in Fig.7 we have reduced the number of plots and enlarged others to improve the quality, in Fig. 7 we also enlarged the plots. However, the plots from flow cytometry analysis presented in the manuscript are copied from Muse Software and in this case we can not improve the smallest letters and numbers in the graphs.

We thank the Reviewer for all suggestions and hope that the revised manuscript is now appropriate for publication.

Sincerely,

Justyna Stefanowicz-Hajduk, Ph.D.

Department of Biology and Pharmaceutical Botany,

Medical University of Gdansk, Gdansk, Poland

Reviewer 3 Report

After careful revision of the manuscript by Stefanowicz-Hajduk entitled ‘In vitro anticancer, antioxidant and phytochemical study on water extract of Kalanchoe daigremontiana Raym.-Hamet & H. Perrier’, I suggest to accept the manuscript for publication after revision of few  minor points, as reported below. Indeed, although some studies on this topic have been already done, the findings reported in the present work could be of interest for researchers in the field.

-Abstract: use only abbreviation or full name for MTT, FRAP, DPPH and so on. The information of both full name and abbreviation is better in the main text;

-I suggest to add a sentence about future perspectives/possible applications, in the bottom of the introduction section. Moreover, it would be better to revise the last sentence (page 2, lines 76-77;

-Page 2, lines 80-82: revise part of the sentence;

-Page 3, line 96: use italics for scientific name of species. Check this in overall text;

-Figure 1: I suggest to revise Fig. 1, in order to report on x axis, the concentration: (i. e. ctrl, 5, 10, 15, 20, 30, 50, 100 and 200 μg/mL) instead of numbers from 1 to 9, as already done for Fig. 2. Revise also Fig. legend accordingly. Moreover, do the values represent the means ± SEM of at least three independent experiments? This information is required also for other Figures. Add in the Figure legend;

-Page 4, lines 123-128: add ‘%’ or ‘μg/mL’ after each number; check this and modify in overall text;

-Page 4, lines 114-117: the authors estimated the effect of K. daigremontiana water extract on the cytostatic action of both paclitaxel and cisplatin in SKOV-3 cells, concluding that no significant changes were observed alone or in combination. How the authors justify these findings? Moreover, why the authors decide to assay different concentration paclitaxel and cisplatin? Wouldn't it have been better to test the same concentrations? Please, explain/ clarify;

-Page 4, lines 127-128: change ‘highest extract concentrations - 200 and 300 μg/mL, respectively 127 (Figure 2)’ by ‘higher extract concentrations (200 μg/mL and 300 μg/mL, respectively), Figure 2’;

-Figure 3. Please provide images with higher resolution. Revise also the arrows (better if smaller), moreover, add the scale bar reporting also the µm of scale bar and the microscope in the Figure legend;

-When statistical analysis was performed, indicate the method used in the Figure legend; and revise the meaning of asterisks: e. g. *p < 0,05; **p < 0,01; *** p < 0,001; **** p < 0,0001 vs. control, depending on analysis and number of asterisks.

-I suggest to add a sentence about future perspectives/possible applications, in the bottom of the discussion section.

Author Response

REVIEWER 3

We would like to thank for critical reading this manuscript and valuable suggestions. We have carefully considered all of the suggestions and made the appropriate corrections.

After careful revision of the manuscript by Stefanowicz-Hajduk entitled ‘In vitro anticancer, antioxidant and phytochemical study on water extract of Kalanchoe daigremontiana Raym.-Hamet & H. Perrier’, I suggest to accept the manuscript for publication after revision of few  minor points, as reported below. Indeed, although some studies on this topic have been already done, the findings reported in the present work could be of interest for researchers in the field.

-Abstract: use only abbreviation or full name for MTT, FRAP, DPPH and so on. The information of both full name and abbreviation is better in the main text;

Response: According to Molecules guidelines for Authors, any “abbreviations in the abstract should be defined the first time they appear in each of three sections: the abstract; the main text; the first figure or table”. 

-I suggest to add a sentence about future perspectives/possible applications, in the bottom of the introduction section. Moreover, it would be better to revise the last sentence (page 2, lines 76-77;

Response: We have corrected this and added a sentence.

-Page 2, lines 80-82: revise part of the sentence;

Response: We have revised this sentence.

-Page 3, line 96: use italics for scientific name of species. Check this in overall text;

Response: We have corrected this.

-Figure 1: I suggest to revise Fig. 1, in order to report on x axis, the concentration: (i. e. ctrl, 5, 10, 15, 20, 30, 50, 100 and 200 μg/mL) instead of numbers from 1 to 9, as already done for Fig. 2. Revise also Fig. legend accordingly. Moreover, do the values represent the means ± SEM of at least three independent experiments? This information is required also for other Figures. Add in the Figure legend;

Response: We have redone the Fig. 1 and added necessary information in the other figure legends.

-Page 4, lines 123-128: add ‘%’ or ‘μg/mL’ after each number; check this and modify in overall text;

Response: We have corrected this.

-Page 4, lines 114-117: the authors estimated the effect of K. daigremontiana water extract on the cytostatic action of both paclitaxel and cisplatin in SKOV-3 cells, concluding that no significant changes were observed alone or in combination. How the authors justify these findings? Moreover, why the authors decide to assay different concentration paclitaxel and cisplatin? Wouldn't it have been better to test the same concentrations? Please, explain/ clarify;

Response: We performed statistical analysis of the results (Anova with post hoc Tuckey’s test, p<0.05) and did not observe significant differences among the results. We decided to tested different concentrations of cytostatics due to different IC50 values of these drugs on SKOV-3 cells found in the literature (according to Smith J.A., Ngo H., Martin M.C., Wolf J.K. An evaluation of cytotoxicity of the taxane and platinum agents combination treatment in a panel of human ovarian carcinoma cell lines. Gynecologic Oncology 2005, 98, 141-145).

-Page 4, lines 127-128: change ‘highest extract concentrations - 200 and 300 μg/mL, respectively 127 (Figure 2)’ by ‘higher extract concentrations (200 μg/mL and 300 μg/mL, respectively), Figure 2’;

Response: We have corrected this.

-Figure 3. Please provide images with higher resolution. Revise also the arrows (better if smaller), moreover, add the scale bar reporting also the µm of scale bar and the microscope in the Figure legend;

Response: We have improved the quality of these images and added the necessary information.

-When statistical analysis was performed, indicate the method used in the Figure legend; and revise the meaning of asterisks: e. g. *p < 0,05; **p < 0,01; *** p < 0,001; **** p < 0,0001 vs. control, depending on analysis and number of asterisks.

Response: We have corrected this.

-I suggest to add a sentence about future perspectives/possible applications, in the bottom of the discussion section.

Response: We have added this sentence.

We thank the Reviewer for all suggestions and hope that the revised manuscript is now appropriate for publication.

Sincerely,

Justyna Stefanowicz-Hajduk, Ph.D.

Department of Biology and Pharmaceutical Botany,

Medical University of Gdansk, Gdansk, Poland